# Comparative Study of Thermal-Oxidative Aging and Salt Solution Aging on Bitumen Performance

**DOI:** 10.3390/ma14051174

**Published:** 2021-03-03

**Authors:** Xuemei Zhang, Inge Hoff

**Affiliations:** Department of Civil and Environmental Engineering, Norwegian University of Science and Technology, Høgskoleringen 7A, 7491 Trondheim, Norway; inge.hoff@ntnu.no

**Keywords:** bitumen, thermal-oxidative aging, salt solution aging, morphology, physical property, low-temperature property, high-temperature property

## Abstract

The aging of bitumen is detrimental to the durability and service life of asphalt pavement. Previous studies found that bitumen was suspected to be aged by not only thermal oxidation but also solution immersion. This research aims to compare the effect of thermal-oxidative aging and salt solution aging on bitumen performance. For this purpose, a thin film oven test (TFOT) and pressure aging vessel aging (PAV) were selected as thermal-oxidative aging, and 10% NaCl aging and 10% CaCl_2_ aging were selected as salt solution aging. The morphology, oxygen content, physical properties, low-temperature properties, and high-temperature properties of bitumen were analysed by employing scanning electron microscopy with an energy dispersive spectrometer (SEM-EDS), physical tests, a bending beam rheometer (BBR), and a dynamic shear rheometer (DSR). Test results show that both thermal-oxidative aging and salt solution aging had similar influencing trends in the oxygen content, physical, low-temperature, and high-temperature properties of bitumen but had different changes in morphology. The aging degrees caused by four kinds of aging methods were obtained based on the summed values of the absolute aging factor of all parameters: PAV > 10% NaCl > TFOT > 10% CaCl_2_. The conclusions could provide a theoretical basis to establish a standard for the solution aging of bitumen.

## 1. Introduction

The aging of asphalt pavement is among the most concerning issues inducing pavement distresses, such as fatigue cracking, thermal cracking, and low-temperature cracking. These distresses would reduce the service life of the pavement and driver’s comfort [1]. The deterioration of asphalt pavement is mainly the result of the aging of bitumen, even though bitumen only accounts for 4–6% of the total mass of the asphalt mixture [2,3]. Based on previous research, the aging of bitumen is attributed to four factors: solar radiation, moisture, time, and temperature [4]. According to the factors influencing the aging of bitumen, different aging modes on bitumen are classified, including thermal-oxidative aging, UV aging, and solution aging [5,6]. Among three aging modes on bitumen, thermal-oxidative aging is the most mature and studied by a large number of researchers, and solution aging is a new formation during recent years [7,8]. Thus, a comparison with thermal-oxidative aging would promote the customization of the specifications of solution aging of bitumen.

The thermal-oxidative aging of bitumen occurs from the construction stage to the end of pavement life. There are two phases for the thermal-oxidative aging of bitumen: short-term aging and long-term aging [9]. Short-term aging presents the thermal aging of bitumen during storage, transportation, and paving, which is normally simulated by the thin film oven test (TFOT) in the laboratory [10]. Long-term aging presents the thermal aging of bitumen during its whole service life, which is simulated by a pressure aging vessel (PAV) in the laboratory [11]. Thermal-oxidative aging of bitumen is principally associated with the oxidative reaction between bitumen and oxygen and the volatilization of light components [12]. Due to the oxidation and the volatilization, the morphological, physical, and rheological properties of bitumen varied [13,14,15,16,17]. These changes in bitumen performance would result in distresses or deteriorations on asphalt pavement.

Although there are no protocols or standards for the solution aging of bitumen, many researchers have found that solution immersion would age the bitumen to some degree. For example, the stiffness of bitumen increased, and oxygen-containing functional groups were produced after solution immersion [18]. Pang [19] also found that solution immersion increased the asphaltene content and complex modulus and reduced the phase angle, which are typical characteristics of the oxidation of bitumen. Additionally, Yang and Wu [20,21] also found that the adhesion and cohesion of bitumen were deteriorated after solution immersion. These changes indicate that the solution immersion would also age bitumen like thermal-oxidative aging but with different mechanisms. Among all solutions based on previous studies, salt solution is among the most used solutions, since salt appears in coastal areas and during winter maintenance, and 10% is used as an estimated average for field condition [22,23,24].

Therefore, this research aims to compare the similarities and differences of the effect of salt solution aging and thermal-oxidative aging on bitumen performance. The salt solution aging, in this case, was chosen as 10% NaCl aging and 10% CaCl_2_ aging. The chosen thermal-oxidative aging methods were TFOT aging and PAV aging. Then, scanning electron microscopy with an energy dispersive spectrometer (SEM-EDS) was used to characterize the morphology and oxygen content of bitumen influenced by aging; penetration, softening point, bending beam rheometer (BBR), and dynamic shear rheometer (DSR) were used to evaluate the physical properties, low-temperature, and high-temperature properties of bitumen before and after different aging methods. The conclusions obtained from this research could provide a theoretical basis to establish a standard for the solution aging of bitumen.

## 2. Materials and Methods

### 2.1. Bitumen

This research adopted a neat bitumen with a penetration grade of 70/100, which is commonly used for constructing asphalt pavement in Norway obtained from Veidekke company. Table 1 lists the basic properties of bitumen.

### 2.2. Aging Methods

There are two kinds of aging modes used in this research: thermal-oxidative aging and salt solution aging. Detailed information about two kinds of aging modes is stated as follows.

#### 2.2.1. Thermal-Oxidative Aging

Thermal-oxidative aging includes thin film oven test (TFOT aging) and pressure aging vessel aging (PAV aging) methods, which simulate the short-term aging and long-term aging of bitumen, respectively [26]. The parameters of thermal-oxidative aging are shown in Table 2.

#### 2.2.2. Salt Solution Aging

There are two kinds of salt solution aging methods of bitumen: 10% NaCl aging and 10% CaCl_2_ aging. The specific parameters of salt solution aging are shown in Table 3. The concentration of salt solution, aging temperature, and aging time were, respectively, selected as 10%, 25 °C, and 90 days based on previous research and the practical situation [27,28]. Therefore, salt solution aging was achieved by immersing bitumen film (28 g) in salt solution (150 mL) for 90 days at 25 °C. To avoid salt solution evaporation, the bitumen immersed in the salt solution was covered by a black plastic bag. After salt solution aging, bitumen samples were cleaned and dried in a fume hood for three days for the following tests. 

### 2.3. Test Methods

#### 2.3.1. Scanning Electron Microscopy (SEM) with Energy Dispersive Spectrometer (EDS)

SEM-EDS was used to characterize the morphology and the oxygen content of bitumen. The bitumen sample (0.01 g) was prepared, then the conductive tape attached to bitumen sample was put on the specimen tube for the following tests. In terms of the bitumen morphology, the bitumen surface was captured using a scanning electron microscope (FlexSEM 1000) at a magnitude of 50 times. The oxygen content indicating the oxidation degree was obtained using an energy dispersive spectrometer (ESD) [29]. It is noted that three samples for each bitumen were tested, and the averaged value of oxygen content was finally used in this research.

#### 2.3.2. Physical Test

To characterize the aging degree of bitumen caused by different aging methods, the physical properties of bitumen should be provided. In this case, penetration (at 25 °C), softening point, and complex viscosity (at 60 °C) of bitumen before and after aging were evaluated according to NS-EN 1426:2015, NS-EN 1427:2015, and NS-EN 13702:2018, respectively. Three, two, and two valid determinations were, respectively, carried out on penetration, softening point, and complex viscosity tests; the average value of each test was used as the test result.

#### 2.3.3. Bending Beam Rheometer (BBR)

The low-temperature properties of bitumen were characterized using a bending beam rheometer according to Norwegian standard NS-EN 14771:2012. Bitumen samples were made to 6.4 ± 0.1 mm high, 12. 7 ± 0.25 mm wide, and 127 ± 5 mm long and tested at −12 and −18 °C. For each type of bitumen, two samples were made for the BBR test. Then, average values of flexural creep stiffness (S) and stress relaxation ability (m-value) at the loading time of 60 s, and the corresponding limited temperature, were calculated and used for evaluating the low-temperature properties of bitumen.

#### 2.3.4. Dynamic Shear Rheometer (DSR)

The high-temperature properties of bitumen were analysed using the dynamic shear rheometer according to NS-EN 14770:2012. The parameters for frequency sweep are listed in Table 4. The master curve of complex shear modulus and phase angle, rutting factor, and failure temperature was carried out to evaluate the high-temperature properties of bitumen. Each bitumen was tested with two replicates, and the average value was finally used.

## 3. Results and Discussions

### 3.1. Morphology and Oxygen Content

Figure 1 presents SEM images of five bitumen samples under original, TFOT, PAV, 10% NaCl, and 10% CaCl_2_ conditions. Some differences can be observed from Figure 1a–c: after TFOT aging, the surface bitumen was rougher with some smaller particles than that of neat bitumen, and these particles were accumulated by large molecules within bitumen; however, the surface of bitumen was cracked and with separated pieces of bitumen after PAV aging. Thus, the thermal oxidation aging increases the proportion of large molecules, resulting in a rough surface with a few cracks [30]. However, from Figure 1a,d,e, salt solution aging induced different changes in the bitumen surface compared to thermal-oxidative aging. Sharp angles of bitumen pieces were observed for salt solution aged bitumen. This result could be attributed to the fact that that salt solution aging would deteriorate the outmost layer of bitumen or lead to residual salt on the bitumen surface, which leads to the loss of bitumen pieces and sharp angles [31]. Thus, bitumen aged by thermal-oxidative aging (TFOT and PAV) showed different phenomena on the morphology of bitumen compared to salt solution aging.

Figure 2 shows the oxygen content of neat bitumen and aged bitumen. The oxygen content in bitumen is one of the most important factors indicating the aging degree of bitumen. The higher oxygen content indicates the more severe aging of bitumen [32]. From Figure 2, it can be seen that the oxygen content of bitumen increased to different extents after four kinds of aging methods were applied. PAV aging led to the most severe aging on bitumen, since the oxygen content of PAV aged bitumen was the highest among four kinds of aged bitumen. The 10% NaCl and 10% CaCl_2_ had similar effects on bitumen aging. TFOT aging has the least effect on bitumen aging, as the oxygen content of TFOT aged bitumen was slightly higher than that of neat bitumen. Therefore, both thermal-oxidative aging and salt solution aging would increase the oxygen content within bitumen, resulting in the more severe aging of bitumen. However, the aging severity of bitumen is dependent on the aging time and aging mechanisms.

### 3.2. Physical Properties

Three parameters were used for indicating the aging degree of bitumen: penetration, softening point, and complex viscosity. Penetration is used to characterize the stiffness or softness of bitumen; a lower value of penetration indicates the higher stiffness of bitumen [33]. The softening point indicates the high-temperature stability of bitumen; a higher softening point value presents greater stability of bitumen at a high temperature [34]. Complex viscosity is applied to measure the resistance to flow of bitumen; a higher value of complex viscosity means a better performance of bitumen to resist flowing [35].

Figure 3 shows the penetration of bitumen with and without aging. It is noted that all aging methods decreased the penetration of bitumen to different extents. For example, PAV aging induced the smallest penetration to 34 dmm, and 10% NaCl also caused a similar reduction in penetration with PAV aging, while TFOT aging and 10% CaCl_2_ decreased the bitumen penetration to 54 and 51 dmm, respectively. These results indicate that both thermal-oxidative aging and salt solution aging could enhance the stiffness of bitumen. Among four aged methods, PAV aging had a similar hardening effect on bitumen with 10% NaCl, and TFOT aging had a similar hardening effect with 10% CaCl_2_. These results indicate that the effect of 10% NaCl aging on bitumen stiffness is equivalent to that of long-term aging, and the effect of 10% CaCl_2_ aging on bitumen stiffness is equivalent to that of TFOT aging.

Figure 4 shows the softening point of bitumen with and without aging. Both thermal-oxidative aging and salt solution aging influence the softening point of bitumen, resulting in a higher value of softening point. However, different aging methods led to different aging degrees of bitumen, arranging the softening point of four kinds of aged bitumen in descending order: PAV aged bitumen, 10% NaCl aged bitumen, TFOT aged bitumen, and 10% CaCl_2_ aged bitumen. These results demonstrate that both thermal-oxidative aging and salt solution aging positively affect the high-temperature stability of bitumen. The effect of salt solution aging on the high-temperature stability of bitumen is similar with that of TFOT aging.

Figure 5 shows the complex viscosity of bitumen at 60 °C with and without aging. It is obvious that all aging methods increased the complex viscosity of bitumen. Among the five kinds of bitumen, PAV aged bitumen has the largest complex viscosity, which is almost four times that of neat bitumen and twice that of the other three aged bitumen. Salt solution aged bitumen showed a slightly higher value than TFOT aged bitumen. The above results demonstrate that thermal-oxidative aging and salt solution would improve bitumen performance to resist flowing, causing an increase in the stiffening effect. Salt solution aging has a similar effect on the performance to resist flowing with TFOT aging.

### 3.3. Low-Temperature Properties

The low-temperature properties of bitumen were evaluated using flexural creep stiffness (S), stress relaxation ability (m-value), and limited temperature and shown in Figure 6 and Figure 7, and Table 5, respectively. Figure 6 shows the flexural creep stiffness (S) of bitumen (at −12 and −18 °C) before and after aging. The higher the value of flexural creep stiffness, the greater the bitumen stiffness. As seen from Figure 6, it was found that bitumen after four kinds of aging methods presented different increments on S values at a low temperature, arranging the S values of four kinds of aged bitumen in descending order: PAV aged bitumen, 10% NaCl aged bitumen, TFOT aged bitumen, and 10% CaCl_2_ aged bitumen. These results indicate that both thermal-oxidative aging and salt solution aging would lead to stiffer bitumen at a low temperature, while salt solution aging has a similar effect on the flexural creep stiffness of bitumen with TFOT aging (short-term aging).

Figure 7 presents the stress relaxation ability of bitumen versus the aging method at −12 and −18 °C. The higher the m-value, the stronger the stress relaxation ability [36]. It could be found that the m-value of bitumen at −12 °C was higher than that at −18 °C, which indicates that bitumen tends to relax more stress at higher temperatures. It is noted that four kinds of aging methods decreased the m-value of bitumen to different extents, arranging the m-value of five kinds of bitumen in descending order: at −12 °C: neat bitumen, TFOT aged bitumen, 10% CaCl_2_ aged bitumen, 10% NaCl aged bitumen, and PAV aged bitumen; at −18 °C: neat bitumen, 10% NaCl aged bitumen, 10% CaCl_2_ aged bitumen, TFOT aged bitumen, and PAV aged bitumen. These results indicate that both thermal-oxidative and salt solution aging would decrease the ability of bitumen to relax stress. PAV aged bitumen had the worst ability to relax stress at two temperatures. TFOT aged bitumen had worse stress relaxation ability at −18 °C and better relaxation ability at −12 °C compared to salt solution aged bitumen. The different orders in the relaxation ability of salt aged bitumen and TFOT aged bitumen at −12 and −18 °C could be attributed different mechanisms of salt solution aging and TFOT aging.

The limited temperature in the BBR test of bitumen with and without aging is shown in Table 5. The limited temperature is determined by the lowest temperature that bitumen fulfils the requirement of S ≤ 300 MPa and m ≥ 0.3 [37]. As seen from Table 5, neat bitumen showed −18 °C limited temperature, while the limited temperature of the other four aged bitumen was −12 °C. These phenomena indicate that neat bitumen could behave well at a lower temperature compared to the other four types of bitumen, while four kinds of aged bitumen have adverse resistance to cracking at a low temperature. Above all, S, m-value, and limited temperature results indicate that both thermal-oxidative aging and salt solution aging would lead to a bad performance of bitumen at a low temperature to different degrees.

### 3.4. High-Temperature Properties

Figure 8 and Figure 9 show the master curve of complex shear modulus (|G*|) and phase angle (δ) of bitumen before and after different aging methods, respectively. According to the time–temperature superposition principle, high frequency corresponds to low temperature, and low frequency is in line with high temperature for bitumen [38]. Based on the definition of complex shear modulus and phase angle, a higher value of complex shear modulus relates to an increase in bitumen stiffness, and a smaller value of phase angle indicates a better elastic response of bitumen [39]. As seen from Figure 8 and Figure 9, it is obvious that phase angle decreased, and the complex shear modulus increased with the increase in frequency. This fact indicates that the elastic response and stiffness of bitumen would increase with the increase in frequency (the decrease in temperature).

The complex shear modulus of bitumen before and after aging is shown in Figure 8. Both thermal-oxidative aging and salt solution aging increased the complex shear modulus of bitumen, especially at a higher frequency (lower temperature). These results indicate that both thermal-oxidative aging (TFOT and PAV) and salt solution aging (10% NaCl and 10% CaCl_2_) would improve the stiffness of bitumen, and the improving degree of four aging methods is arranged in descending order: PAV aging, TFOT aging, 10% NaCl aging, and 10% CaCl_2_ aging. It is worth noting that the difference between aged bitumen and neat bitumen in complex shear modulus increased over frequency, which indicates that the aging process has a more significant effect on the stiffness of bitumen at a high frequency (low temperature) than at a low frequency (high temperature).

Considering the effect of aging on phase angle seen from Figure 9, bitumen showed different reactions on phase angle compared to complex shear modulus. TFOT, 10% NaCl, and 10% CaCl_2_ aged bitumen presented little changes in phase angle at a low frequency (high temperature) compared to neat bitumen, while they presented large differences in phase angle at a high frequency (low temperature). The above results demonstrate that salt solution aging has a similar effect on bitumen viscoelasticity to TFOT aging. These three aging methods primarily influence the low-temperature properties of bitumen, resulting in a better elastic response of bitumen at a high loading frequency (low temperature). Meanwhile, TFOT aging improved the elastic response of bitumen more than salt solution aging at a high frequency (low temperature). However, a better elastic response of bitumen at a low temperature would cause more risks of cracks, which is regarded as a deterioration of the low-temperature properties of bitumen. Therefore, both TFOT and salt solution aging thus deteriorated the low-temperature properties of bitumen, while TFOT aged bitumen presented slightly worse low-temperature properties of bitumen compared to salt solution aging. These results are consistent with the results from the BBR test. In addition, PAV aging had a dramatic effect on the phase angle of bitumen, resulting in a totally smaller phase angle of bitumen over frequency compared to neat bitumen. This phenomenon demonstrates that PAV aging leads to a better elastic response of bitumen, regardless of frequency or temperature. Additionally, PAV aged bitumen showed a similar phase angle with TFOT, 10% NaCl, and 10% CaCl_2_ aged bitumen at a high frequency. This fact means that PAV aging has a similar effect on the elastic response of bitumen to the other three aging methods at a high frequency.

Figure 10 shows the rutting factor of bitumen with and without aging. The rutting factor |G*|/sinδ is a parameter indicating the ability of bitumen to resist rutting at a high temperature. The higher the rutting factor, the greater the ability of bitumen to resist rutting [40]. As seen from Figure 10, the aging increased the rutting factor of bitumen, PAV aged bitumen had the largest rutting factor, and TFOT and salt solution aging had similar effects on the rutting factor of bitumen, whereas neat bitumen had the smallest value of rutting factor. These results indicate that both thermal-oxidative and salt solution aged bitumen have a better ability to resist rutting compared to neat bitumen, arranging the rutting resistance of four kinds of aged bitumen in descending order: PAV aged bitumen, TFOT aged bitumen, 10% NaCl aged bitumen, and 10% CaCl_2_ aged bitumen.

Additionally, the failure temperature was calculated as the critical temperature at which the rutting factor (at 10 rad/s) was equal to 1 kPa and is shown in Table 6. The failure temperature of bitumen determines the temperature at which the bitumen fails, so it is regarded as a factor indicating high-temperature stability [41]. Aged bitumen showed a higher failure temperature than neat bitumen, which means that aged bitumen is less prone to failure and the aging has a positive effect on the stability of bitumen at a high temperature. Comparing four kinds of aging methods, PAV aging caused the most significant effect on the failure temperature of bitumen, followed by TFOT aging, and NaCl aging; 10% CaCl_2_ had the smallest effect on failure temperature.

### 3.5. Comparison of Bitumen Aging Degree Caused by Thermal-Oxidative Aging and Salt Solution Aging

The above test results showed that salt solution would also lead to bitumen aging. In order to systematically compare thermal-oxidative aging and salt solution aging, the aging degree of bitumen caused by different aging methods was characterized by the aging factor of the main property parameters [42]. In this research, the aging factor (AF) of oxygen content, penetration, softening point, and complex viscosity; the average value of AF of flexural creep stiffness and stress relaxation ability (at −12 and −18 °C); complex shear modulus |G*|; phase angle δ; and rutting factor (at 30, 40, 50, 60, 70, and 80 °C) were calculated according to Formula 1 and shown in Table 7. The positive and negative values of AF indicate that the parameters increase or decrease after different aging methods.
(1)$$\mathrm{Aging}\;\mathrm{factor}\;(\mathrm{AF})=\frac{{\mathrm{Parameter}}_{\mathrm{after}\;\mathrm{aging}}-{\mathrm{Parameter}}_{\mathrm{neat}\;\mathrm{bitumen}}}{{\mathrm{Parameter}}_{\mathrm{neat}\;\mathrm{bitumen}}}$$
where ${\mathrm{Parameter}}_{\mathrm{after}\;\mathrm{aging}}$ indicates the property parameter of bitumen after aging, and ${\mathrm{Parameter}}_{\mathrm{neat}\;\mathrm{bitumen}}$ indicates the parameter of neat bitumen. 

As seen from Table 7, salt solution aging had the same varying trend with thermal-oxidative aging, leading to negative AF values of penetration, m-value, phase angle, and positive AF values of the rest of the parameters. However, PAV aging resulted in the highest value of the sum of absolute values of all AF, followed by 10% NaCl, and TFOT aging; 10% CaCl_2_ had the smallest value of the sum of absolute values of all AF. These results indicate that PAV aging leads to the most severe aging on bitumen, followed by 10% NaCl, and TFOT aging; 10% CaCl_2_ causes the least aging on bitumen. Meanwhile, the sum of absolute values of all AF of salt solution aged bitumen is close to that of TFOT aged bitumen, which indicates that the influence of salt solution aging on bitumen performance is comparable to that of TFOT aging (short-term aging).

## 4. Conclusions

This paper compared the effect of thermal-oxidative aging and salt solution aging on the bitumen performance of laboratory samples.

Bitumen showed the same varying trends under salt solution aging and thermal-oxidative aging in terms of oxygen element, physical properties and low-temperature properties, and high-temperature properties, resulting in the aging of bitumen. However, different effects on the morphology of bitumen caused by salt solution aging compared to thermal-oxidative aging were observed, i.e., salt solution aging led to sharp angles of bitumen pieces, while thermal-oxidative aging induced the rough surface of bitumen with certain cracks. 

The aging degrees caused by thermal-oxidative aging and salt solution aging were different, arranging the aging degree caused by four kinds of aging methods in descending order: PAV aging, 10% NaCl aging, TFOT aging, and 10% CaCl_2_ aging. These differences in morphology and aging degree of bitumen caused by thermal-oxidative and salt solution aging are attributed to the different conditions and mechanisms of aging methods. 

The aging effect of salt solution aging on bitumen performance is almost equivalent to that of TFOT aging, except for morphology.

This research revealed that salt solution aging had a similar aging effect on bitumen performance with TFOT aging. The experimental design and comparison results can provide a theoretical basis for the establishment of bitumen solution aging standards. However, the results in this research are only valid for neat bitumen, and it should be followed by further investigations on the effect of different aging methods on modified bitumen and the mechanism of salt solution aging on bitumen in future studies.

## Figures and Tables

**Figure 1 materials-14-01174-f001:**
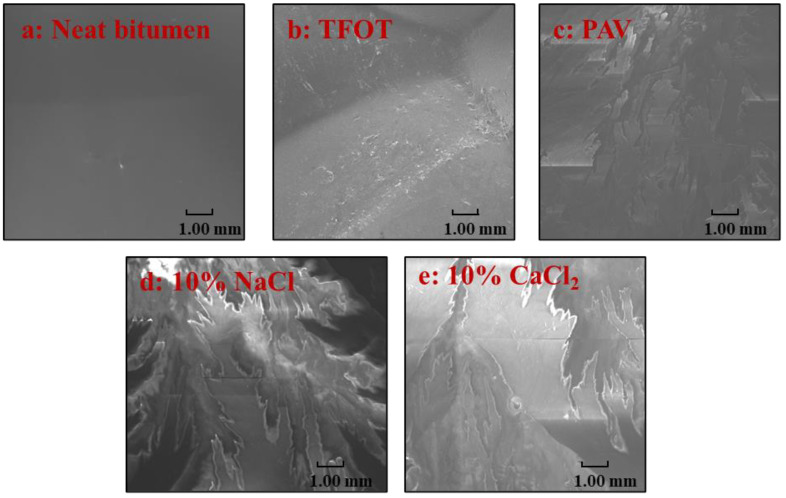
SEM images of bitumen before and aging: (**a**) Neat bitumen; (**b**) TFOT; (**c**) PAV; (**d**) 10% NaCl and (**e**) 10% CaCl_2_.

**Figure 2 materials-14-01174-f002:**
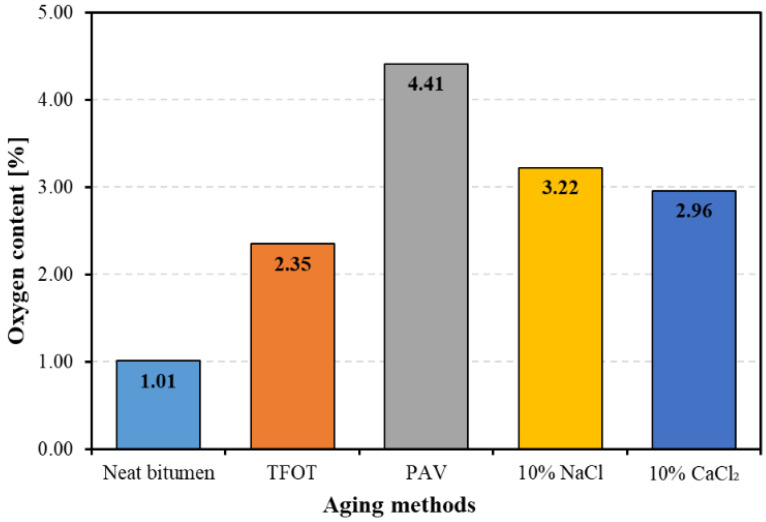
The oxygen content of bitumen before and after aging.

**Figure 3 materials-14-01174-f003:**
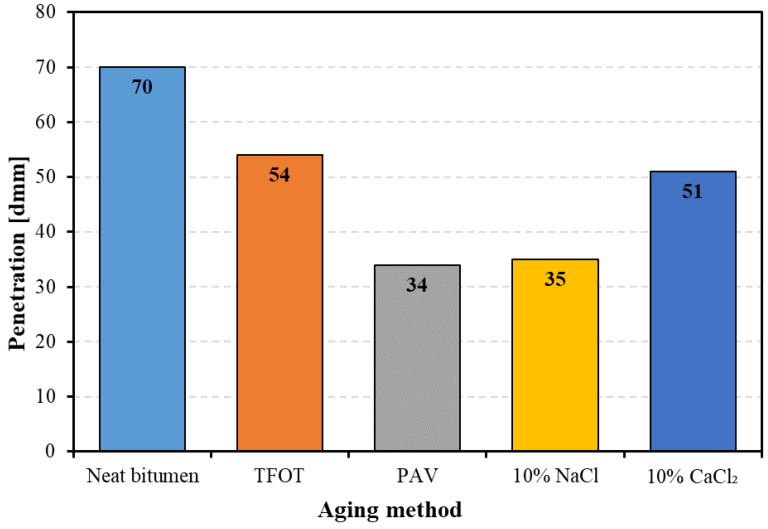
The penetration of bitumen before and after aging.

**Figure 4 materials-14-01174-f004:**
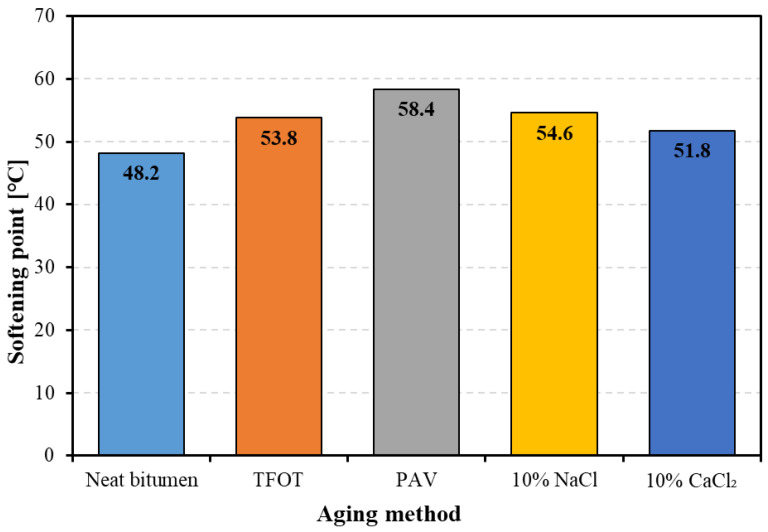
The softening point of bitumen before and after aging.

**Figure 5 materials-14-01174-f005:**
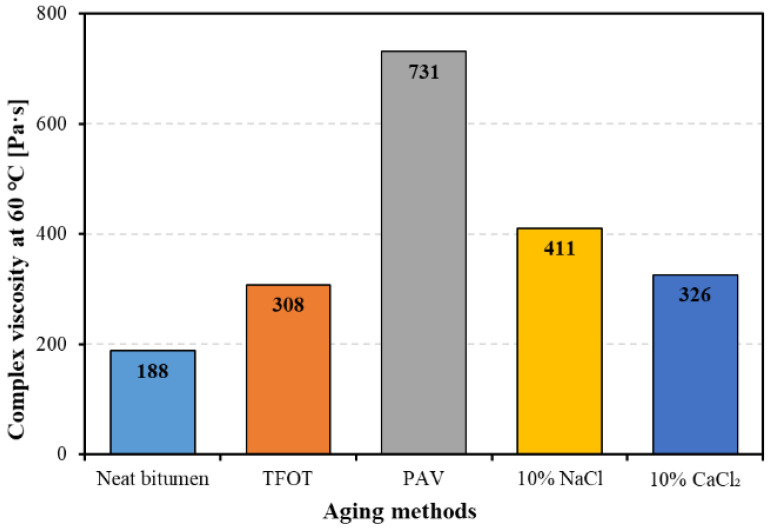
The complex viscosity of bitumen before and after aging.

**Figure 6 materials-14-01174-f006:**
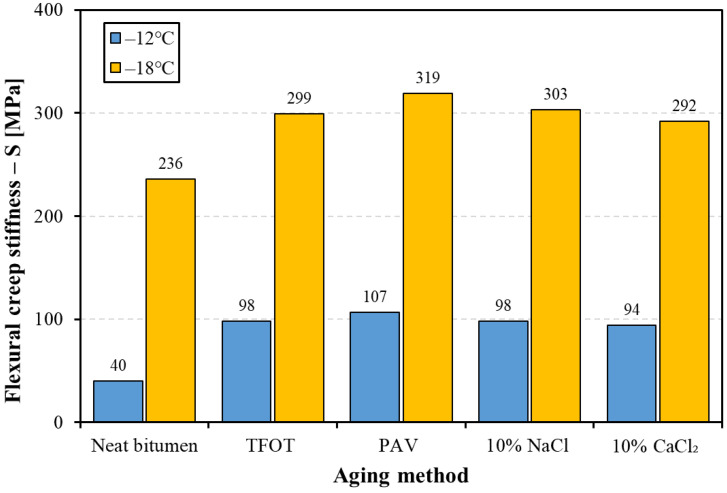
The flexural creep stiffness of bitumen before and after aging.

**Figure 7 materials-14-01174-f007:**
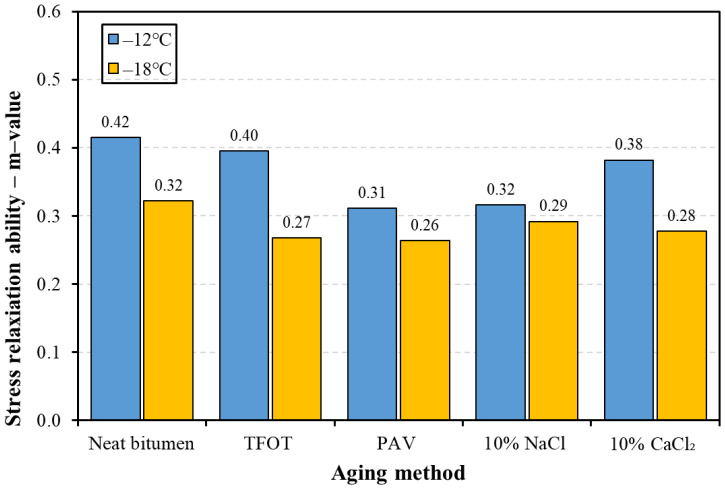
The stress relaxation ability of bitumen before and after aging.

**Figure 8 materials-14-01174-f008:**
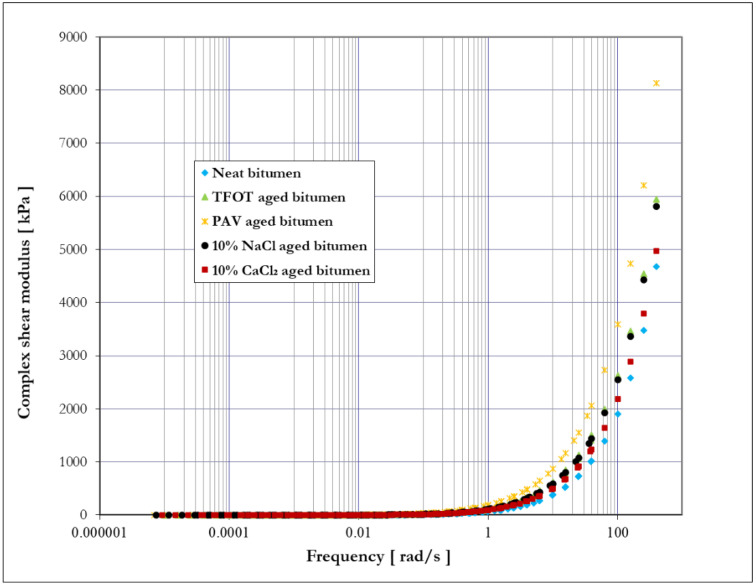
Master curve—complex shear modulus versus loading frequency.

**Figure 9 materials-14-01174-f009:**
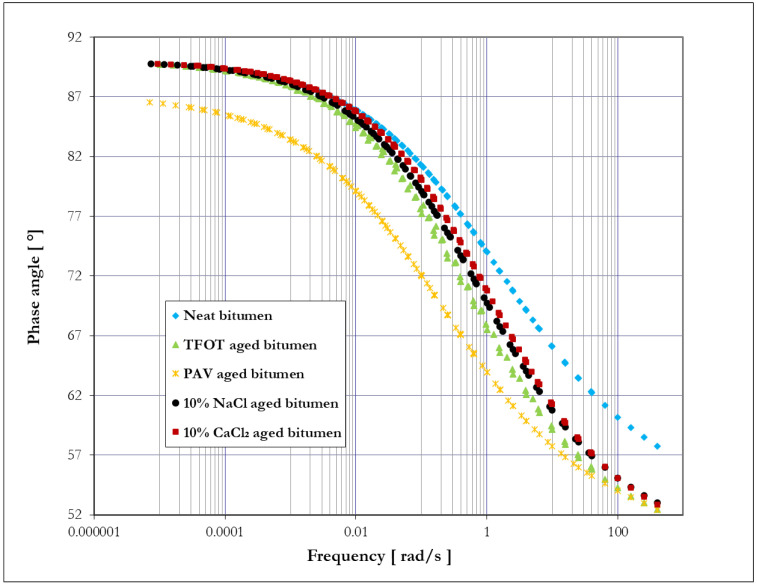
Master curve—phase angle versus loading frequency.

**Figure 10 materials-14-01174-f010:**
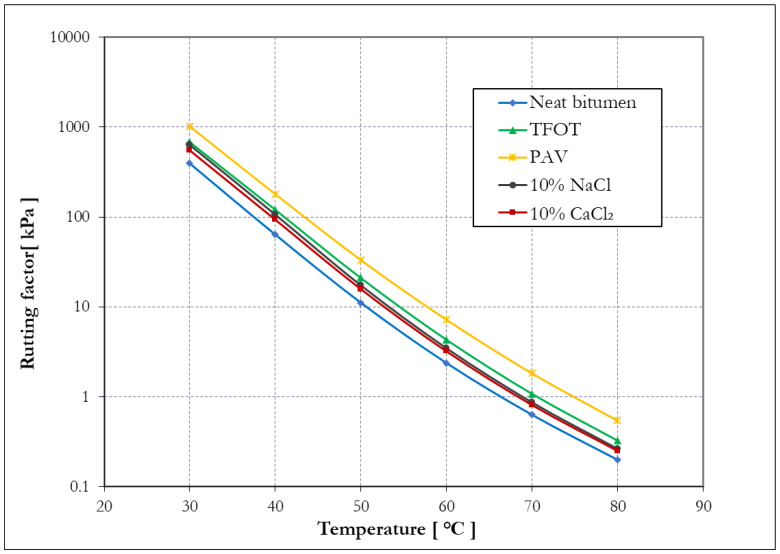
The rutting factor of bitumen before and after aging.

**Table 1 materials-14-01174-t001:** Basic properties of neat bitumen.

Property	Penetration (25 °C)	Softening Point	Viscosity (60 °C)	Limited Temperature in BBR Test
Neat bitumen	70 dmm	48.2 °C	188 Pa∙s	−18 °C
Test standard	NS-EN 1426:2015	NS-EN 1427:2015	NS-EN 13702:2018	Performance grade [25]

**Table 2 materials-14-01174-t002:** Parameters of thermal-oxidative aging.

Parameter	TFOT Aging	PAV Aging
Temperature	163 °C	100 °C
Aging time	5 h	20 h
Bitumen mass	50 g	50 g
Standard	NS-EN 12607-2:2014	NS-EN 14769:2012

**Table 3 materials-14-01174-t003:** Parameters for salt solution aging.

Parameter	10% NaCl/CaCl_2_ Aging
Temperature	25 °C
Aging time	90 days
Bitumen mass	28 g
Diameter of container	190 mm

**Table 4 materials-14-01174-t004:** Parameter settings.

Setting	Sweep Frequency	Strain Value	Test Temperature (°C)
Value	0.1–400 rad/s	1%	30, 40, 50, 60, 70, 80

**Table 5 materials-14-01174-t005:** The limited temperature of five kinds of bitumen.

Bitumen Type	Neat Bitumen	TFOT Aged Bitumen	PAV Aged Bitumen	10% NaCl Aged Bitumen	10% CaCl_2_ Aged Bitumen
Limited temperature [°C]	−18	−12	−12	−12	−12

**Table 6 materials-14-01174-t006:** The failure temperature of five kinds of bitumen.

Bitumen Type	Neat Bitumen	TFOT Aged Bitumen	PAV Aged Bitumen	10% NaCl Aged Bitumen	10% CaCl_2_ Aged Bitumen
Failure temperature (°C)	65	70	74	69	68

**Table 7 materials-14-01174-t007:** The aging factor of four kinds of aging methods.

Aging Method	TFOT	PAV	10% NaCl	10% CaCl_2_
AF-Oxygen content	1.33	3.37	2.19	1.93
AF-Penetration	−0.23	−0.51	−0.50	−0.27
AF-Softening point	0.12	0.21	0.13	0.07
AF-Complex viscosity	2.00	9.05	3.71	2.29
$\bar{\mathrm{AF}}$-S	0.86	1.01	0.87	0.24
$\bar{\mathrm{AF}}$-m-value	−0.11	−0.21	−0.17	−0.11
$\bar{\mathrm{AF}}$-Complex shear modulus |G*|	0.75	1.75	0.49	0.35
$\bar{\mathrm{AF}}$-Phase angle δ	−0.05	−0.10	−0.03	−0.02
$\bar{\mathrm{AF}}$-Rutting factor |G*|/sin δ	0.78	1.84	0.50	0.36
Sum of absolute value of all AF	6.23	18.05	8.59	5.64

## Data Availability

Data sharing is not applicable to this article.

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
