# Peer review of "Comparative Study of Thermal-Oxidative Aging and Salt Solution Aging on Bitumen Performance"

_materials, 2021, doi:10.3390/ma14051174_

Round 1

Reviewer 1 Report

Dear authors,

it was an interesting and practical paper I have now read. Nevertheless I have several recommendations, unclarities or questions.

  • I assume the two salt solutions were selected because they are typical, but I have not found on what the 10% solution was based. This would be good just to include so the read knows.
  • For the conditioning, in case of TFOT or PAV we know what the time period of the test simulates – more or less. But how is it with the 90 days for the salt dilutions? Can they be related to some time period what they would on a pavement simulate?
  • I understand you chose one bitumen type, but would the conclusion be valid for 160/220 as well? Most probably yes. But what about PMBs? That is pity that at least a PMB representative was not selected as well.

I put further comments and proposals to the attached file. I also recommend to have a careful English proof-reading since some sentences are in my opinion not English-correct.

Reviewer 2 Report

Review:

 Comparative study of thermal-oxidative aging and salt solution aging on bitumen performance

Abstract:

Well‐written, concise, clear

Subject matter is original, press-worthy, of major general interest.

Introduction:

Hypothesis and purpose of study are clearly and concisely presented

Materials and methods

Adequately written, although writing could be polished; minor typos/grammar/ punctuation errors

  • Description of procedures needs minor clarification (clearly remediable)

Description of procedures unclear; would be difficult for others to reproduce study by reading the article, although with rewriting, the deficit could be remedied.

Please provide more information concerning the SEM EDS measurements.

Results and discussion

Statistical significance of findings not stated

Remark:

 3.1. Morphology and oxygen content

However, from Figure A, D, and E, salt solution aging induced different changes in bitumen compared to thermal-oxidative aging.

Remark: use image 1a, 1b.....

Conclusion:

Statements and conclusions are presented but need minor revision to correlate more clearly with data and link with goals.

  • Study implications and/or limitations are presented but are missing a point(s). Please add limitations and possible recommendations for future work.

Reviewer 3 Report

The paper is an interesting work on bitumen performance evaluation comparing different aging processes. In particular, the authors studied the aging response caused by salt solution and thermal radiation. The study of bitumen salt solution aging is an innovative idea to analyse the performance of the binder.

The paper is not set to be reviewed. It very difficult to write a review in this condition! The authors did not insert the numeration at side of the paper pages.

I suggest the authors to check the English.

In the abstract, second and third lines contains repetitions: “aged” …. “aging”. The term “erosion” is not adapted, and I suggest the authors to replace it with a term similar to “aging”.

The term “lab” in the “Introduction” section (17th and 18th lines after the “Introduction” title) is not correct. I suggest the authors to write “laboratory” instead of “lab”.

Page 7, 4th line of the subsection 3.3. “Low-temperature property”: the sentence “The higher value of flexural creep stiffness, stiffer of the bitumen.” is not clear.

The test methods need to be adequately described. In particular, the salt solution aging process is not clear. It is difficult to understand how the authors performed the salt solution aging. The authors did not explain well the process. In the abstract the authors wrote (5th line of the abstract) “…10 % NaCl AND 10% CaCl2 were selected as salt solution aging”, but in the “Introduction” section they wrote (18th line from the top of page 2) “…in 10% NaCl OR 10% CaCl2 for 90 days”. At 39th line from the top of page 2, the authors wrote again “…10 % NaCl AND 10% CaCl2 for 90 days”. It is not clear to me how runs the salt solution aging. The samples were subjected to 10 % NaCl salt solution and then to 10% CaCl2 one, or some samples were subjected only to 10 % NaCl salt solution and some only to 10% CaCl2 one? It is not clear also the duration of the process: 90 days for each salt solution or for both ones. It is not clear also why the authors use 25 °C to perform the salt solution aging. I suggest the authors to wrote in the paper the reason to set the test temperature to 25 °C.

Moreover, I suggest the authors to wrote in the paper how many samples were used for each test.

Table1, page 2: penetration result needs to be reported (as wrote in the standard) in tenths of millimetres (not in millimetres) and rounded to the nearest integer. The authors need to write the year of the standard “NS-EN 1426”, how similarly they wrote for the other ones.

I suggest the authors to modify the all penetration values wrote in paper: i.e. in subsection 3.2 “Physical property” at page 5, in figure 3 at page 6 and in table 6 at page 12.

Table 2: I suggest the authors to modify the format of the paper, without insert half table in one page and other half one in the second page.

Page 3: I suggest the authors to explain the data in Table 3. What is amplitude gamma equal to 1%? What is the temperature interval equal to 10 °C?

Page 5: the authors wrote (3rd line from the top of the subsection 3.2. “Physical property”) “…higher value of penetration indicates higher stiffness of bitumen…”. I suggest the authors to replace “higher” with “lower”.

Page 6, subsection 3.2 “Physical property”: I suggest the authors to modify all softening point values as indicated in the standard NS-EN 1427:2015. The softening point, for softening points below or equal to 80 °C, need to be express to the nearest 0,2 °C; instead, the softening point, for softening points above 80 °C, need to be express to the nearest 0,5 °C.

Between the 4th and the 1st lines from the bottom of the page 6 the authors wrote “The above results demonstrate that thermal-oxidative aging and salt solution would improve bitumen performance to resist flowing, and salt solution aging has a similar effect on the performance to resist flowing with short-term aging.” I suggest the authors to insert “and then the stiffening effect is higher” after “…salt solution would improve bitumen performance to resist flowing…” because it seems that bitumen improves performance (but this is not verifiable in this context).

Page 7, 7th and 8th lines of the subsection 3.3. “Low-temperature property”: the sentence “Arranging the S values of bitumen after four aging methods in descending order: PAV > 10% NaCl > TFOT > 10% CaCl2.” is not corrected. I suggest the authors to use a sentence like what they used at page 6 (4th and 5th lines of the first paragraph) “Arranging the softening point of four kinds of aged bitumen in descending order: PAV aged bitumen, 10% NaCl aged bitumen, TFOT aged bitumen, 10% CaCl2 aged bitumen.”.

Page 8, 5th line from the top of the page: it is not clear to me the meaning of the words “Two orders”? I suggest the authors to re-write the second and the third sentences of the page 8. The second sentence is “The reduction degree at -18 in descending order is PAV, TFOT, CaCl2, and NaCl, and the descending order at -12 °C is PAV, NaCl, CaCl2, and TFOT.”, while the third one is “Two orders provide that PAV aged bitumen had the worst ability to relax stress at two temperatures, TFOT aged bitumen has worse stress relaxation ability at -18 °C and better relaxation ability at -12 °C compared to two kinds of solution aging methods, while NaCl and CaCl2 aged bitumen have better ability to relax stress at lower temperature but worse ability at higher temperature.

I think that the bitumen does not contain water because it is a hydrophobic material, then the residual NaCl and CaCl2 in bitumen cannot act as deicing agents. I suggest the authors to modify the sentence “The better relaxation ability of NaCl and CaCl2 aged bitumen at lower temperature could be interpreted that residual NaCl and CaCl2 in bitumen as deicing agents could improve the low-temperature properties of bitumen.” (at the end of the first paragraph at the page 8).

Page 9, 7th line from the top of the page: “Figure 3” it is not corrected. Modify please.

Page 9, between 9th line and 11th line: the sentence “This fact indicates that more elastic components of bitumen would exist, and a better deformation resistance would be performed at a higher frequency (lower temperature).” it is not corrected because the authors stated at the end of the page 8 that “…results indicate that both thermal-oxidative aging and salt solution aging would lead to a bad performance of bitumen at low temperature…”. Please review and correct your affirmations. Moreover, the fact that “phase angle decreased, and the complex modulus increased with the increasing of frequency” does not mean that “more elastic components of bitumen would exist”. The aging process stiffer the bitumen and the complex modulus increased a little bit (with respect the neat bitumen). At low temperatures (high frequency) the complex modulus of PAV aged bitumen is very similar to the neat bitumen one and the difference in the phase angle values (between PAV aged and neat bitumen) is not explainable with a presence of more elastic “components”. If the authors stiff a bitumen with an aging process, they obtain a more viscous behaviour. I suggest the authors to clarify and enlarge they opinion on the results of DSR tests.

Page 9, 17th and 18th lines from the top of the page: the sentence “….and the improving degree is arranged as follows: PAV > 10% NaCl > TFOT > 10% CaCl2.” is not corrected. I suggest the authors to use a sentence like what they used at page 6 (4th and 5th lines of the first paragraph) “Arranging the softening point of four kinds of aged bitumen in descending order: PAV aged bitumen, 10% NaCl aged bitumen, TFOT aged bitumen, 10% CaCl2 aged bitumen.”.

I suggest the authors to modify the affirmations at 16th and 20th lines from the top of the page 9: the sentences “both thermal-oxidative aging (TFOT and PAV) and salt solution aging (10% NaCl and 10% CaCl2) would improve the ability of bitumen to resist deformation” and “the aging processes have a more significant effect on the deformation resistance of bitumen at high-temperature but a little influence on that at low temperature” are not clear to me. The aging processes stiff the neat bitumen and did not improve the ability of bitumen to resist deformation. Otherwise, why all people did not perform a long-aging process (i.e. PAV) on the bitumen before to lay and compact the asphalt mixtures during the construction of the roads?

I suggest the authors to clarify the sentences at page 10 (between 1st line and 16th line from the top of the page). The PAV procedure simulate the asphalt binder aging that occurs during 5 - 10 years of in-service hot mix asphalt pavements (long term aging). It is obvious that the aging process make the bitumen more prone to thermal cracking, brittle behaviour with low temperature and more viscous behaviour with high temperature. What it is the meaning of the sentence “the PAV aging would improve the high-temperature properties and deteriorate the low-temperature properties of bitumen”?

Page 10, 6th line from the bottom of the page: “Figure 6” it is not corrected. Modify please.

Page 10, 3rd and 4th lines from the bottom of the page: the meaning of sentence “These results indicate that aged bitumen has a better ability to resist rutting compared to neat bitumen…” is obvious (similar to discover America). The PAV aging process became the bitumen more viscous, more resistant to rutting and less prone to thermal susceptivity. But the aging process also make bitumen more prone to cracking and with lower cohesion and adhesion forces.

Page 10, 1st and 2nd lines from the bottom of the page: the sentence “….and the order of rutting resistance of bitumen is PAV aged bitumen > TFOT aged bitumen > 10% NaCl aged bitumen > 10% CaCl2 aged bitumen.” is not corrected. I suggest the authors to use a sentence like what they used at page 6 (4th and 5th lines of the first paragraph) “Arranging the softening point of four kinds of aged bitumen in descending order: PAV aged bitumen, 10% NaCl aged bitumen, TFOT aged bitumen, 10% CaCl2 aged bitumen.”.

The subsection 3.5. “The aging degree of bitumen caused by thermal oxidative aging and salt solution aging” is an arbitrary evaluation invented by the authors, without any objective reason. I suggest the authors to remove this sub-section from the paper. What is the meaning of “positive” and “negative” “change”? Positive and negative for what? The sums wrote in Table 6 do not have any physical meaning. The authors sum together: millimetres, degree Celsius, percentage, megapascal, complex modulus, etc.

The authors, reporting the differences in the Table 6, need to write the unit of measurements at the side of the different parameters (oxygen content, penetration, ecc.). A difference between two physical measurements have the same unit of measurement of the initial values (making a subtraction between two values with the same unit of measurements, the result do not has the unit of measurement).  Moreover, penetration and softening point values are also badly written because did not follow the general rules indicated in the standards.

In the conclusions section, I suggest the authors to control and modify the format (the bulleted list is at left with respect to the text of paper).

Generally, is not possible to insert in a paper the conclusion in a bulleted list, because a scientific paper is not a summary. Please delete the bulleted list and re-write the conclusions. I suggest the authors to remove: the sentence connected to the first bullet of the list because did not generate any new conclusion; the sentence connected to the fourth bullet because the meaning is not corrected (“…PAV aging significantly improved the elasticity of bitumen at low frequency.”)

The sentence of the last bullet “Integrating the effect of all aging parameters on bitumen, arranging the aging degree in descending order: PAV > 10% NaCl > TFOT > 10% CaCl2.” is not corrected. I suggest the authors to use a sentence like what they used at page 6 (4th and 5th lines of the first paragraph) “Arranging the softening point of four kinds of aged bitumen in descending order: PAV aged bitumen, 10% NaCl aged bitumen, TFOT aged bitumen, 10% CaCl2 aged bitumen.”.

Reviewer 4 Report

The overall goal of this manuscript is unclear. The main purpose of the paper is to compare different aging methods: PAV, TFOT and aging caused by two salt solution. Why?

What is purpose of the aging solution? To simulate a road under water? To simulate the effect of winter maintenance? There is a lack of precise goal and scope.

"The salt solution aging, in this case, was conducted by immersing bitumen in 10% NaCl or 10% CaCl2 for 90 days". Why? Why 90 days and 10%?The comment percentage is 30%. This sentence is also repeated twice in the text.

SEM methods. There is no description of any protocol for sampling. How the samples have been produced? How much materials have been used? 1grams? 100 grams? SEM is also for conductive samples, how authors overcome this barrier?

Physical test. No standards followed?

DSR. Same comment as above.

0.1-400 rad/s1%30-80 °C10 °C. why?

“This result could be interpreted that salt solution aging would deteriorate the outmost layer of bitumen, which leads to the erosion of the edge or top layer of bitumen [26], leading to the loss of bitumen pieces and sharp angles.” Maybe the sharp edge is the salt? The salt stays, unless it does evaporate or disappear.

Figure 1A is very distorted. Any reason for that? I would have expected something as in https://www.researchgate.net/figure/SEM-morphology-of-bitumen-samples-a-B-b-PMB3-c-PMB5-and-d-PMB7_fig3_338627118

The oxygen content is in percentage but it is not clear how this percentage is calculated. Where does this number come from?

“These results demonstrate that both thermal-oxidative aging and salt solution aging positively affect the high-temperature stability of bitumen to a different extent”. Which extent exactly?

“The better relaxation ability of NaCl and CaCl2aged bitumen at lower temperature could be interpreted that residual NaCl and CaCl2in bitumen as deicing agents could improve the low-temperature properties of bitumen.” By looking at the phase diagraph of  CaCl2 solutions, it is possible to see that for 10%CaCl2 solution between -12/-18, the phase is solution+ice. So the above assumption would not be correct.

The eutectic composition of the sodium chloride -water system is 23 percent NaCl and 77 percent H20 by weight, which freezes at about -21°C. Some evidence suggest that anti-icing operations should not be conducted (using liquid, prewetted, or dry salt) when the pavement temperature is at or below about -9.5°C (15°F). Some highway agencies also believe that it is not practical to use salt below -9°C (15°F) for general snow and ice control operations, at least not without calcium chloride. This experience has convinced them that salt’s action is too slow at these lower temperatures. An inspection of Figure 16 shows that the phase diagrams of NaCl and CaCl2 are not too dissimilar in the temperature range of 0°C (32°F) down to -10°C (15°F) or even down to about -15°C (5°F). Thus, the two chemical brines have about the same solidification (freezing) characteristics in this temperature range. The fact that calcium chloride has a much lower eutectic temperature than sodium chloride is not of importance for anti-icing operations. Source: FHWA

Indeed, it is mostly used BBR in combination with DSR. It is not clear why authors have not conducted DSR on low temperatures. The information that the tests give are different and complementary.  “the 4-mm DSR generally shows lower limiting temperature values than BBR and in some cases, difference between the two test methods can be over 10 °C (Lu et al., 2017)” or “The Dynamic Shear Rheometer (DSR) is used for estimating the high PG while the Bending Beam Rheometer (BBR) is the reference method for obtaining the low PG. This is associated to a number of drawbacks both from a practical testing viewpoint (different specimen preparation) and because of the costs required to purchase two devices. Therefore, if the DSR can be used to characterize asphalt binder at low, intermediate and high temperature, this would be beneficial for practitioners on a testing routine basis” (Losa et al., 2017). At this point the reviewer stops doing literature review and invite the authors to really motivate their choices.

How is the rutting factor calculated? Where does it come from? There is only a reference that does not describe any rutting factor. Ref 19 has a plot with (G*/sinδ). Is this what authors are trying to describe?

If I look at page 10-11 for example, the references go from number 19 to 37, and then 28. Why are they not in order?

“Aged bitumen showed a higher failure temperature than neat bitumen, which means aged bitumen is hard to fail and the aging has a positive effect on bitumen performance at high temperature”. I would expect that aging causes the pavement to be more susceptible to cracking. What do authors mean by saying performance? Rutting? Fatigue? Cracking?

Table 6 is absolutely incomprehensible. Where do these numbers come from? How these values are calculated? “by the change rate of all parameters of bitumen compared to neat bitumen”. Which change? Which rate? Is it exactly a copy-paste of the reference paper 28?

The conclusions have a different layout than the rest of the text.

“This research found that salt solution aging has asimilar aging effect on bitumen performance with thermal-oxidative aging with different mechanisms, which means the influence of salt solution aging should be paid the same attention as thermal-oxidative aging.” From the results of the manuscript, it seems that PAV is 1) a standardized method and 2)it is the one that has more aging effect so why one should analyzed water solution effect unless the road is not immersed in water?

Round 2

Reviewer 1 Report

Dear authors,

the revised version is fine to me. I found only 2 or 3 small errors and provided one comment to the change rate. It is all in atttached file.

Author Response

Thank you very much for your efforts and valuable comments to improve the quality of paper. The revised parts (round 2) were marked with Blue in the revised manuscript.

Point 1: Delete and start the sentence with "Detailed...."; MPa (I know word is often changing this to Mpa); I am not sure if it is better to use "bad" or "adverse"; Shall we not add that it is complex shear modulus?

Response 1: According to your comments, “And the” was deleted on line 83; “Mpa” was revised as “MPa” on line 243; “bad” was revised as “adverse” on line 247; “complex modulus” has been revised as “complex shear modulus” on line 132, 254, 258, 259, 261, 264, 265, 271, 277, Figure 8, and Table 7.

Point 2:  Can be the change rate parameter somehow interpreted as aging factor? I am fine with change rate, just if the "aging factor" would not better stress what interprets the rate.

Response 2: Thank you very much for the suggestions. Yes, the change rate (CR) can be interpreted as aging factor in this research. The authors have revised the CR to “aging factor (AF)” for better and direct understanding. The revised parts were located on line 22, 325, 326, 327, 330, 333, 336, 337, 338, 339, 342, and Table 7.

Reviewer 4 Report

No further comments.

Author Response

Thank you very much for your efforts and professional comments.